# Evaluation of Aerobic Propagation of Yeasts as Additional Step in Production Process of Corn Ethanol

Matheus Ribeiro Barbosa Oliveira [1,*], Rafael Soares Douradinho [1], Pietro Sica [2], Layna Amorim Mota [3], Alana Uchôa Pinto [1], Tamires Marques Faria [1] and Antonio Sampaio Baptista [1,*]

1 Department of Agri-Food Industry, Food and Nutrition, College of Agriculture "Luiz de Queiroz", University of São Paulo, Padua Dias Avenue, 11, Piracicaba 13148-900, SP, Brazil; rafael.douradinho@usp.br (R.S.D.); alanauchoap@usp.br (A.U.P.); tamiresfaria@usp.br (T.M.F.)
2 Department of Plant and Environmental Sciences, University of Copenhagen, Thorvaldsenvej, 40, 1821 Frederiksberg, Denmark; pietro@plen.ku.dk
3 Center for Nuclear Energy in Agriculture, University of São Paulo, Centenário Avenue, 303, Piracicaba 13416-000, SP, Brazil; layna.amorim@usp.br
* Correspondence: mathribeiro@usp.br (M.R.B.O.); asbaptis@usp.br (A.S.B.)

**Abstract:** Yeast is one of the co-products of ethanol plants, which can be used as a nutritional supplement in animal feed due to its high protein content. Given the importance of yeast contribution to the nutritional properties of DDG (dried distillers' grains), the aim of this study was to assess how different levels of aeration affect the biomass production and the quality of yeast providing new insights into yeast production, offering an alternative source of income for the corn ethanol industry. For this purpose, yeasts were grown in a fed-batch process, and different concentrations of aeration in the medium were tested, namely 0.5, 1.0, and 1.5 volume of air per volume of wort per minute (v v$^{-1}$ min$^{-1}$). At the end of the cellular biomass production process, yeasts grown with 0.5 (v v$^{-1}$ min$^{-1}$) aeration in the reactor showed higher biomass formation (19.86 g L$^{-1}$), cellular yield (g g$^{-1}$), and a lower formation of succinic acid (0.70 g L$^{-1}$) and acetic acid (0.11 g L$^{-1}$). Aeration influenced an increase of 1.0% in the protein content in yeast. In conclusion, lower levels of aeration in the yeast production process enables more efficient sugar utilization for biomass formation and is a potential strategy to increase the protein content and the commercial value of DDG.

**Keywords:** aeration; biomass production; cell growth; dried distillers' grains; nutrition; oxidative stress; oxygen; protein

## 1. Introduction

The world population is continuously increasing, posing a challenge to provide essential proteins and nutrients for both humans and animals [1]. By 2050, the global population could rise to 9.1 billion. Consequently, the demand for water and food, particularly protein, will increase [2]. Proteins are necessary for metabolic function and serve as a source of nitrogen for humans and animals to build the structural and functional units necessary for life. However, humans and animals cannot synthesize certain essential amino acids, so they consume meat or protein sources to meet these needs [3].

Yeast is one of the co-products of ethanol plants, which can be used as a nutritional supplement in animal feed due to its high protein content (39–54%) [4]. *Saccharomyces cerevisiae* yeast has been extensively grown, both under anaerobic conditions to produce ethanol and under aerobic conditions to produce baker's yeast and biomass for use in food and feed [5]. This versatility arises because *Saccharomyces cerevisiae* has the ability to grow under both anaerobic and aerobic conditions, which gives it a unique physiology [6]. Under anaerobic conditions, pyruvate is reduced to ethanol with the release of $CO_2$. Theoretically, up to 51.1 g of ethanol and 48.9 g of $CO_2$ can be obtained per 100 g of glucose metabolized. Additionally, in the glycolysis stage, two molecules of ATP (adenosine triphosphate) are

produced per glucose molecule. On the other hand, under aerobic conditions and low glucose concentration (0.1 g $L^{-1}$), pyruvate is oxidized to produce two molecules of $CO_2$ and ATP. The oxidation of sugars under aerobic conditions represents a more efficient energy pathway than anaerobiosis, yielding around 30 molecules of ATP per molecule of glucose [7].

Yeast is a valuable animal feed supplement due to its high protein content. Distillers' dried grains (DDG), a by-product of corn ethanol production plants, are used as animal feed because a significant portion of its protein content is derived from yeast after the fermentation process [8]. DDG serve as an additional source of income for corn ethanol plants, and its value is directly linked to its protein content. Therefore, increasing the protein content in DDG could enhance its overall value [9]. In corn ethanol production, yeast cells are not recycled and are instead incorporated into DDG [10]. Therefore, each fermentation cycle requires the introduction of a new inoculum. Consequently, yeast multiplication through aerobiosis offers a method to generate a substantial amount of yeast biomass to initiate each new fermentation cycle and enhance the protein content of DDG with minimal sugar consumption. For this reason, the aim of this study was to assess how different levels of aeration affect the biomass production and the quality of yeast for animal feed purposes. This study provides new insights into yeast production, offering an alternative source of income for the corn ethanol industry.

## 2. Results

### 2.1. Cell Viability

The yeast subjected to treatments T1 (cell growth at 0.5 volume of air per volume of wort* per minute), T2 (cell growth at 1.0 volume of air per volume of wort* per minute), and T3 (cell growth at 1.5 volume of air per volume of wort* per minute) started cell growth with 80% cell viability. However, during the first two hours of growth, a reduction in cell viability was observed, reaching values of 69.0%, 68.0%, and 70.0% for treatments T1, T2, and T3, respectively. Thus, after 2 h of growth, the cell viability of yeast subjected to treatments T1, T2, and T3 exhibited an upward trend throughout the cell growth period until the end of cell growth, in 16 h. At this time, the yeast cells subjected to treatments T1, T2, and T3 showed, respectively, a cell viability of 79.0%, 78.0%, and 80.0%. Therefore, no significant differences were observed in the cell viability of yeast exposed to T1, T2, and T3 treatments during cell growth (Figure 1).

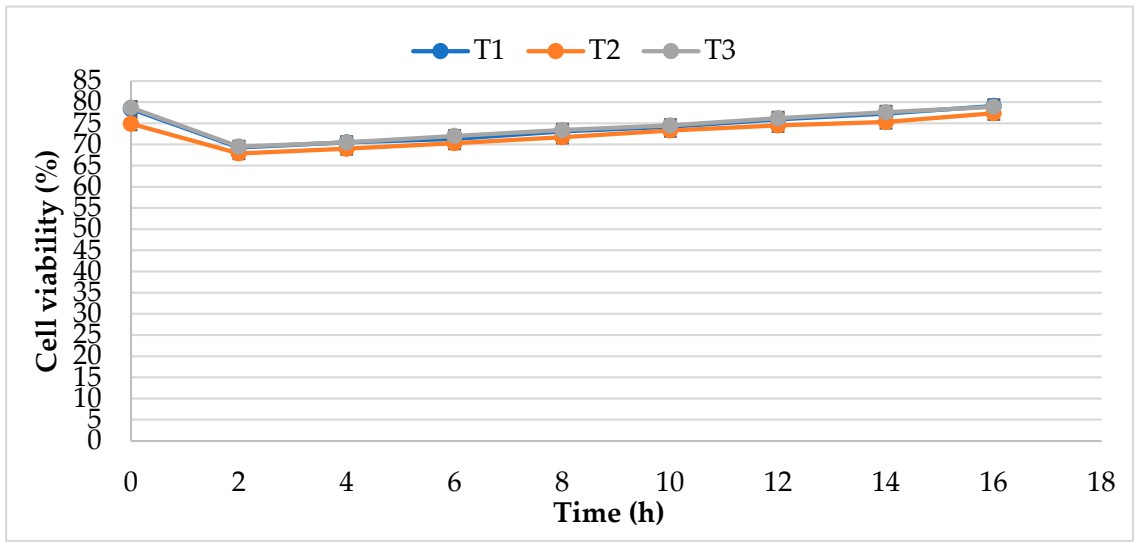

**Figure 1.** Cellular viability of yeast subjected to treatments T1 (cell growth at 0.5 volume of air per volume of wort per minute), T2 (cell growth at 1.0 volume of air per volume of wort per minute), and T3 (cell growth at 1.5 volume of air per volume of wort per minute) over 16 h of cell growth.

## 2.2. Sugar Assimilation

The yeast subjected to treatments T1, T2, and T3 was cultivated in medium initially containing 0.1 g L$^{-1}$ of sugar. During the 16 h of cell growth, 79.3 g of sugar was dispensed into the reactor, in order to maintain the concentration of the medium at 0.1 g L$^{-1}$ of the total reducing sugar. As a result, at the end of cell growth, yeast grown under the conditions of treatments T1, T2, and T3 consumed all the sugar from the substrate in the reactor. (Table 1) In this way, yeasts used all available sugar as a carbon source for several metabolic processes, including the production of cellular biomass (mainly), organic acids, glycerol, trehalose, and ethanol.

**Table 1.** Parameters observed at beginning and end of cell growth of yeasts subjected to treatments T1 (cell growth at 0.5 volume of air per volume of wort per minute), T2 (cell growth at 1.0 volume of air per volume of wort per minute), and T3 (cell growth at 1.5 volume of air per volume of wort per minute).

| Parameters | | Treatments | | |
|---|---|---|---|---|
| | Initial | T1 Final | T2 Final | T3 Final |
| Cellular biomass (g L$^{-1}$) | 9.00 [d] ($\pm$0.10) | 19.86 [a] ($\pm$0.30) | 18.07 [b] ($\pm$0.40) | 14.60 [c] ($\pm$0.30) |
| Sugar content in the reactor (g L$^{-1}$) | 0.10 [b] ($\pm$0.02) | 0.00 [a] ($\pm$0.00) | 0.00 [a] ($\pm$0.00) | 0.00 [a] ($\pm$0.00) |
| Alcohol content (v v$^{-1}$) | 0.00 [d] ($\pm$0.00) | 1.08 [b] ($\pm$0.10) | 1.35 [a] ($\pm$0.10) | 0.53 [c] ($\pm$0.10) |
| Cellular yield (Y x/s) (g g$^{-1}$) | 0.00 [d] ($\pm$0.00) | 0.25 [a] ($\pm$0.00) | 0.23 [b] ($\pm$0.00) | 0.18 [c] ($\pm$0.00) |
| Ethanol yield (Y p/s) (g g$^{-1}$) | 0.00 [b] ($\pm$0.00) | 0.01 [a] ($\pm$0.00) | 0.02 [a] ($\pm$0.00) | 0.01 [a] ($\pm$0.00) |
| Succinic acid (g L$^{-1}$) | 0.00 [d] ($\pm$0.00) | 0.70 [c] ($\pm$0.05) | 0.96 [b] ($\pm$0.06) | 1.16 [a] ($\pm$0.05) |
| Acetic acid (g L$^{-1}$) | 0.00 [d] ($\pm$0.00) | 0.11 [c] ($\pm$0.05) | 0.33 [b] ($\pm$0.05) | 0.84 [a] ($\pm$0.06) |
| Lactic acid (g L$^{-1}$) | 0.00 [a] ($\pm$0.00) | 0.00 [a] ($\pm$0.00) | 0.00 [a] ($\pm$0.00) | 0.00 [a] ($\pm$0.00) |
| Trehalose (g 100 mL$^{-1}$) | 9.09 [d] ($\pm$0.00) | 11.11 [c] ($\pm$0.20) | 11.67 [b] ($\pm$0.20) | 13.41 [a] ($\pm$0.25) |
| Glycerol (g L$^{-1}$) | 0.00 [c] ($\pm$0.00) | 4.08 [b] ($\pm$0.12) | 4.12 [b] ($\pm$0.15) | 4.88 [a] ($\pm$0.17) |
| Protein (%) | 52.50 [b] ($\pm$0.00) | 53.00 [a] ($\pm$0.20) | 53.20 [a] ($\pm$0.20) | 53.30 [a] ($\pm$0.30) |

The same medium was used for the three treatments, so the baseline values were the same for all treatments. The means of the same parameter followed by different letters, on the same line, differ statistically from each other using the Tukey test at a 5% significance level ($p \leq 0.05$).

## 2.3. Cellular Biomass

To initiate cell growth, 9 g L$^{-1}$ of dry mass of Saccharomyces cerevisiae, strain Thermosacc Dry® from Lallemand®, located in Piracicaba, Brazil, was used for each treatment. Upon the completion of cell growth, 19.86 g L$^{-1}$, 18.07 g L$^{-1}$, and 14.60 g L$^{-1}$ (dry mass) of yeasts subjected to treatments T1, T2, and T3, respectively, were produced. These data reveal that the T1 treatment provided an increase in biomass production of 9.9% in relation to T2 and 36.0% in relation to T3. Meanwhile, T2 provided an increase in biomass production of 23.8% compared to T3. This shows that the lower aeration applied to the reactor favored the higher formation of cellular biomass. Furthermore, it showed a tendency towards reduction in cell biomass as aeration in the reactor increased. In other words, the use of 0.5 volume of air per volume of wort* per minute provided the best result (Table 1).

## 2.4. Alcohol Content

During the first 8 h of cell growth, there was no production of ethanol by yeast exposed to T1 and T2 treatments, while yeast subjected to T3 treatment began ethanol synthesis only after 10 h of cell growth (Figure 2). Therefore, until the start of ethanol production, yeasts subjected to treatments T1, T2, and T3 used all the sugar for the aerobic pathway.

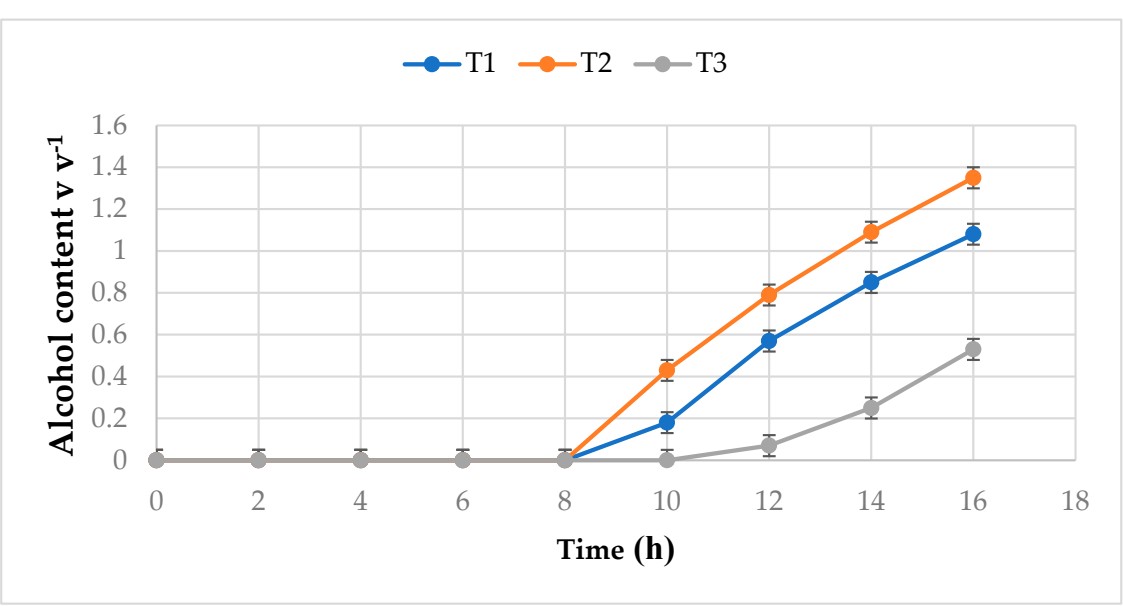

**Figure 2.** Alcohol content produced by yeast subjected to treatments T1 (cell growth at 0.5 volume of air per volume of wort* per minute), T2 (cell growth at 1.0 volume of air per volume of wort* per minute), and T3 (cell growth at 1.5 volume of air per volume of wort* per minute) over 16 h of cell growth.

As soon as the yeast subjected to treatments T1, T2, and T3 began to produce ethanol, a progressive increase in alcohol content was observed in the following hours. At the end of 16 h of cell growth, there was a significant difference in the alcohol content produced between the yeasts subjected to treatments T1, T2, and T3, producing 1.08 v v$^{-1}$, 1.35 v v$^{-1}$, and 0.53 v v$^{-1}$ of ethanol, respectively. This means that the T2 treatment provided an increase in alcohol content of 60.7% compared to T3 and 20.0% compared to T1. Meanwhile, T1 provided an increase in alcohol content of 50.9% compared to T3. These results show that the T3 treatment produced a lower alcohol content, as the presence of higher aeration in the cellular growth medium resulted in less sugar being used to produce ethanol (Figure 2).

*2.5. Cellular Yield (Y x/s)*

At the end of the cell growth process, it was evident that the application of treatments T1, T2, and T3 resulted in statistically significant differences in cell yield (Y x/s). The values corresponding to these cell yields (Y x/s) were 0.25, 0.23, and 0.18 for treatments T1, T2, and T3, respectively. These results indicate that the amount of air injected into the reactor influences cell production and yield. Therefore, the T1 treatment applied the lowest aeration in the reactor and as a consequence provided the highest cell yield. Meanwhile, the T3 treatment that employed greater aeration in the reactor reduced cell yield. Therefore, these findings highlight the sensitivity of cell production to specific aeration conditions, highlighting the importance of optimizing these conditions to maximize cell yield in the context of the growth process (Table 1).

*2.6. Organic Acids*

At the end of the cell biomass production process, 0.40 g L$^{-1}$, 0.65 g L$^{-1}$, and 0.86 g L$^{-1}$ of succinic acid were produced by yeasts subjected to treatments T1, T2, and T3, respectively. Thus, yeasts subjected to T1 treatment produced a 38.5% lower amount of succinic acid compared to T2 and 53.5% compared to T3. Meanwhile, yeast exposed to T2 treatment produced a 24.4% lower amount of succinic acid compared to T3. Additionally, 0.11 g L$^{-1}$, 0.33 g L$^{-1}$, and 0.98 g L$^{-1}$ of acetic acid were also produced by yeast subjected to treatments T1, T2, and T3, respectively. Therefore, the yeasts subjected to T1 treatment produced 66.6% less acetic acid compared to T2 and 88.8% compared to T3. Meanwhile, yeast exposed

to T2 treatment produced 66.3% less acetic acid compared to T3. However, there was no production of lactic acid by yeast subjected to treatments T1, T2, and T3. These results show that using less aeration in the reactor leads to a reduction in organic acid production by yeasts, allowing them to utilize the available sugar more efficiently for cellular biomass production (Table 1).

### 2.7. Trehalose

The yeast started cell growth with 9.09 g 100 mL$^{-1}$ of trehalose and at the end of cell growth 11.11 g 100 mL$^{-1}$, 11.67 g 100 mL$^{-1}$, and 13.41 g 100 mL$^{-1}$ of trehalose by yeast subjected to treatments T1, T2, and T3, respectively. Therefore, yeast subjected to treatment T1 produced 4.0% less trehalose compared to T2 and 13.7% less compared to T3. On the other hand, yeast exposed to treatment T2 produced 24.4% less trehalose compared to T3. These results show that reduced aeration in the reactor is associated with lower trehalose production by yeast. This suggests that when receiving less aeration, yeasts direct their energy and resources towards the production of cellular biomass instead of using the available sugar for the synthesis of trehalose, which is generally associated with cellular protection (Table 1).

### 2.8. Glycerol

At the end of the cell biomass production process, 4.08 g L$^{-1}$, 4.12 g L$^{-1}$, and 4.88 g L$^{-1}$ of glycerol were produced by yeast cultivated under the conditions of treatments T1, T2, and T3, respectively. These data reveal that the T3 treatment provided an increase in glycerol production of 19.6% compared to T1 and 18.4% compared to T2. Meanwhile, there was no statistical difference in relation to glycerol production between treatments T1 and T2 (Table 1). These findings show that increased aeration in the reactor is related to higher glycerol production by yeast. This shows that when receiving a high volume of aeration, besides the optimum, yeasts starting to use part of their energy and resources towards cellular protection instead of using the available sugar for biomass synthesis.

### 2.9. Protein

At the end of cell biomass production, 53.0%, 53.2%, and 53.3% of protein were found in yeast cultivated under the conditions of treatments T1, T2, and T3, respectively. This means an increase of 1.0%, 1.3%, and 1.5% in protein was found in yeast subjected to treatments T1, T2, and T3. However, there was no significant difference between treatments in terms of protein production. These results indicate that yeast cultivated under the growth conditions of treatments T1, T2, and T3 can be utilized to compose products used for animal supplementation, as there will be no significant loss in protein content (Table 1).

## 3. Discussion

The results obtained in the present study indicate that it was possible to carry out cell growth in a fed batch with Thermosacc Dry® yeast, using corn hydrolyzate, aiming for its application in ethanol production. To achieve this objective, yeast was grown in medium supplemented with nutrients, and the medium feed was adjusted so that the sugar concentration remained at 0.1 g L$^{-1}$. This is because the limiting substrate concentration in the bioreactor will control the specific rates of substrate consumption, growth, and product formation due to common phenomena such as the activation, inhibition, induction, and repression of enzymes. Thus, it is advantageous to manipulate the feed rate so that the limiting substrate concentration in the bioreactor remains constant [11–13]. Thus, when oxygen is sufficient and the sugar concentration is equal to or less than 0.1 g L$^{-1}$, small-quantity or no ethanol is produced, and the yeast follows oxidative phosphorylation, with oxygen as the final electron acceptor (production of biomass). However, when there is an increase in sugar in the medium, the Crabtree effect occurs, in which glucose represses the synthesis of several respiratory enzymes, and ethanol is the main product formed [14–16].

To optimize glucose-sensitive fed-batch aerobically grown yeast cultures in terms of cell yield, it is necessary to analyze growth behavior in the range where metabolism switches from full respiration to aerobic fermentation [17]. The maximum yield coefficient (Y x/s) of oxidative yeast growth reaches the range of 0.50 g of biomass/g of glucose. However, if sugar is supplied in excess, the formation of ethanol, resulting in a decrease in cell yield, cannot be avoided even in the presence of excess oxygen. Likewise, insufficient sugar supply decreases cell yield [16]. There are several factors such as substrate concentration, nutrient availability, the medium aeration rate, and the formation of undesirable products, among others [18]. In this case, it is ideal to reduce variation in control variables to optimize cell yield. One way to optimize yield is to control the feed rate. There are several studies that have investigated effective approaches, such as adaptive control, model predictive control, fuzzy control, and control based on artificial neural networks, for controlling the feed rate during a fed-batch process [12]. In this present study, the feeding model was used according to Equation (1) [19]. At the end of the cell growth process, yeast subjected to treatments T1, T2, and T3 showed a cell yield (Y x/s) lower than the maximum yield described in the literature [20]. This observation can be attributed to overflow metabolism which is characterized by the rapid transition to aerobic fermentation as soon as respiratory capacity becomes saturated [21]. This made it evident that an online measuring device to detect overflow metabolism is necessary to handle low glucose concentrations. However, there is currently no commercial device that can measure glucose concentration.

Oxygen plays an essential role in the aerobic growth of yeast, because as the concentration of yeast cells increases, the demand for oxygen also increases proportionally. If this demand exceeds the oxygen transfer capacity of the fermenter, the oxygen transfer rate becomes the limiting factor for cell growth. Therefore, it is crucial to ensure that the amount of oxygen injected into the reactor is adjusted as cells increase in the culture medium, in order to avoid limitations in cell growth. [18,22]. In this regard, the present study tested three aeration conditions in yeast cell culture, these being 0.5, 1.0, and 1.5 volume of air per volume of wort* per minute (v $v^{-1}$ $min^{-1}$). The results obtained reveal that the T1 treatment provided an increase in biomass production of 36.0% in relation to T2 and 9.9% in relation to T3. This shows that the lower aeration applied to the reactor favored the greater formation of cellular biomass. Furthermore, a tendency towards reduction in cell biomass production was observed as the aeration rate in the reactor increased, this reduction being particularly evident in the T3 treatment. This reduction in biomass production in response to increased aeration can be attributed to possible yeast degeneration caused by the formation of highly reactive oxygen species (ROS). ROS have the ability to attack essential cellular components such as DNA, lipids, sugars, and proteins, which can lead to cell death [23,24].

In the cell growth carried out in this study, yeasts subjected to treatments T1, T2, and T3 produced ethanol, even in the presence of aeration, at certain moments of the process. The yeasts subjected to the T1 and T2 treatments produced ethanol after 8 h and the yeasts subjected to the T3 treatments after 10 h of cell growth. The variation in alcohol production times by yeast is due to the fact that biomass production at that time was lower than expected production, according to Equation (1), and used to control the specific flow rate for feeding the peristaltic pumps. Therefore, there was an increase in the sugar concentration in the medium. This observation can be attributed to overflow metabolism that attributes aerobic fermentation to the saturation of a limited respiratory capacity (maximum ATP production by mitochondria is achieved) leading to an aerobic overflow reaction at the pyruvate level. This overflow effect is characterized by the rapid transition to fermentation once respiratory capacity becomes saturated and is distinct from the long-term effects related to respiration repression [21,25]. At the end of the cellular biomass production process, the comparison between yeasts grown under the conditions of treatments T1 and T2 revealed that when using lower aeration in the reactor, yeasts produce fewer organic acids and can thus use the available sugar more efficiently to produce more cellular biomass. Postma and colleagues [26] observed that acetate formation occurred

simultaneously with ethanol formation and a decrease in cellular biomass formation. This is attributed to the effect of accumulated and unmetabolized weak acids, which act as uncouplers, affecting energy metabolism in yeast, due to their diffusion through membranes with proton gradients. By dissipating the proton gradient across the plasma membrane, more ATP is hydrolyzed by ATPase to maintain homeostasis. This requires enhanced mitochondrial ATP synthesis and consequently more intense respiration [26–28]. However, yeasts subjected to conditions of higher aeration in the medium (treatment T3) produced more succinic and acetic acids than yeasts cultivated under the conditions of treatments T1 and T2. This happened because the yeasts were cultivated in high-aeration conditions. Under high-concentration aeration conditions, oxygen can cause yeast degeneration through the formation of highly reactive oxygen species that attack cellular compounds including DNA, lipids, sugars, and proteins, which can result in cell death [24]. Furthermore, organic acids may also have contributed to the formation of these reactive oxygen species. Abbott and colleagues [23] observed that weak organic acids increase oxidative stress in aerobic yeast cultures. In the context of this study, it is possible that a synergistic effect occurred between organic acids and high oxygen concentration in promoting the formation of reactive oxygen species.

During aerobic growth, glycerol plays an important role as an osmoregulator in cells and can increase cellular "turgor pressure" to facilitate sprout formation [29]. Glycerol is formed when there is an increase in glucose concentration, where respiration is repressed, reducing the organism's ability to reoxidize excessive amounts of NADH formed in biosynthetic reactions [30]. Thus, in the present study, 4.08 g $L^{-1}$, 4.12 g $L^{-1}$, and 4.88 g $L^{-1}$ of glycerol were produced by yeast cultivated under the conditions of treatments T1, T2 and T3, respectively. These results demonstrate that under the conditions of high aeration in the reactor (1.5 v $v^{-1}$ $min^{-1}$), there is a higher production of glycerol by yeast. This is attributed to the fact that glycerol is formed during cell growth under stress conditions caused by the high concentration of aeration, where glycerol functions as an efficient osmolyte protecting the cell against lysis [31,32].

Trehalose is a non-reducing disaccharide composed of two glucose residues linked by an $\alpha$-1,1 glycosidic bond. The intracellular concentration of trehalose plays an important role in the ability of yeast to tolerate adverse environmental conditions [33]. During aerobic growth, a metabolic function of trehalose synthesis is to restrict the flow of glucose to glycolysis, inhibiting the initial steps of the glycolytic pathway [29]. Trehalose has been shown to be an essential factor in membrane stabilization, being proposed as a protector against severe dehydration and desiccation stress. Trehalose also plays a protective role in osmoregulation, protecting cells under the conditions of nutrient limitation and starvation. Additionally, high levels of trehalose in yeast are associated with greater osmotolerance, thermotolerance, and ethanol tolerance [34]. Therefore, the results of the present work showed that the reduction in aeration in the reactor is associated with a lower production of trehalose by yeast. This suggests that when receiving less aeration, yeasts direct their energy and resources towards the production of cellular biomass instead of using the available sugar for the synthesis of trehalose, which is associated with cellular protection against adverse environmental conditions [33].

In this study, a significant increase in trehalose production was observed by yeast subjected to T1, T2, and T3 treatments compared to their initial levels of trehalose. This increase in trehalose production by cells is of great importance as it contributes to improved long-term preservation properties, as well as resulting in improved fermentation capacity. This is particularly desirable in yeasts used for ethanol production, where the ability to tolerate adverse conditions and maintain cell viability over time is crucial to the efficiency of the process [35,36].

The commercialization of dried distillers' grains (DDG) is fundamental to the viability of corn-based ethanol production plants. Therefore, factors that influence the quality of DDG have a direct impact on the economics of ethanol production [37]. A variable that affects market value is the variation in the composition of DDG [8]. The main criteria to

consider include crude protein, fat, and fiber content. Fiber content can be subdivided into acid detergent fiber (ADF) or neutral detergent fiber (NDF), along with total carbohydrate and hemicellulose content. All of these factors contribute to the quality of DDG as animal feed. Protein is the most valuable component in the animal diet, and its DDG content can vary from 27% to 33% on a dry basis [38].

The composition of DDG attracts great interest in the area of animal nutrition due to its high levels of energy, protein, and phosphorus. The protein present in DDG is mainly derived from yeast after the fermentation process [8]. At the end of the cell biomass production process, protein levels of 53.0%, 53.2%, and 53.3% were found in yeast cultivated under the conditions of treatments T1, T2, and T3, respectively. This represents an increase of 1.0%, 1.3%, and 1.5% in the protein content found in yeast subjected to treatments T1, T2, and T3. These results indicate that cell multiplication by aerobiosis is a possible strategy to increase the amount of protein in DDG, since the yeast produced contains a higher protein content and can be used both for alcoholic fermentation and incorporated directly into DDG. This occurs because aeration is an important factor in obtaining a greater formation of viable cellular biomass during cell growth, as air injection promotes a greater conversion of the carbon source into biomass [39,40].

In the plant, fermenters are utilized for fermentation processes and can vary in size, with 200 $m^3$ fermenters for cellular growth being common. However, it is important to note that the usable liquid volume typically represents only 75% of the total volume due to expansion caused by dispersed air bubbles and foam formation [39]. Considering the data found in this study, it is possible to estimate the amount of yeast produced and the total reducing sugar used in the process in a 150 $m^3$ fermenter. Thus, from 79.3 g of total reducing sugar, 19.86 g $L^{-1}$ of yeast were produced. Therefore, in a 150 $m^3$ fermenter, it would be possible to produce 2979 kg of yeast using 11.9 tons of total reducing sugar.

On the other hand, in the cellular production process by anaerobiosis, which is currently used, an estimate can be made using theoretical growth yield data of 0.05 and a total reducing sugar concentration in the wort of 120 g $L^{-1}$ [41]. Thus, theoretically, in a 150 $m^3$ fermenter, it would be possible to produce 900 kg of yeast using 18 tons of total reducing sugar. Therefore, comparing the two yeast production processes, it is possible to conclude that the anaerobiosis growth process would consume 6.1 tons of total reducing sugar more than the aerobic growth process. This surplus amount of total reducing sugar could be used to produce ethanol or yeast. In the case of sugar being destined for alcohol production, it would be possible to produce up to 3950 L of ethanol. However, if the surplus amount of total reducing sugar were used for biomass production, the aerobic process would produce 3606 kg more yeast than anaerobiosis, meaning four times more yeast would be produced. However, if the aerobic cellular production process of this study were optimized, it would be possible to produce twice as much yeast, further intensifying the difference between producing yeast by aerobic and anaerobic processes.

Therefore, if the aerobic production process were employed for biomass production instead of the current anaerobic process, the same 18 tons of sugar used to produce 900 kg of yeast for fermentation process could also generate an additional 3606 kg of yeast. Considering the yeast's protein content of 53%, this would result in 1911 kg of protein available for incorporation into DDG at the plant. By incorporating this protein into DDG, currently ranging from 27% to 33% protein content, it would be possible to produce DDG with a higher protein content, resulting in a co-product with enhanced market value. In corn ethanol production, DDG is a co-product that the market price varies according to its protein content. Therefore, the higher the protein content, the higher the market price of DDG. Thus, by adopting an aerobic cell growth stage, the plant can optimize the allocation of surplus sugar based on the prevailing prices of each commodity throughout the year.

## 4. Materials and Methods

### 4.1. Preparation of Wort

The wort used was derived from the hydrolysis of corn, which was subjected to milling in a hammer mill, Tramontina® (Porto Alegre, Brazil), model TRE40, to obtain fragments with a particle size smaller than 2 mm, using a sieve. The hydrolysis process began with the cooking of the particulate corn suspension to a temperature of 86 °C. For this, 1 kg of corn particulate was added to every 2 L of water with adjusted pH 5.8. Then, in a temperature of 86 °C, α-amylase enzyme was added at a concentration of 0.025% m/m$^{-1}$ of corn particulate, and the mixture was kept under constant stirring for 150 min (liquefaction). After the corn starch liquefaction was completed, the system was cooled until the temperature stabilized at 65 °C. At this point, the pH of the mixture was adjusted to 5.0. Under these conditions, the amyloglucosidase enzyme was added at 0.056% m/m$^{-1}$ of corn particulate, and the mixture was kept under constant stirring for 150 min (saccharification). To remove the particles, the material was centrifuged at 3925× *g* for 20 min using a refrigerated centrifuge, Thermo Scientific® (Waltham, MA, USA), model Sorvall ST 40R [42]. Then, the corn hydrolysate was sieved with a 0.037 mm mesh sieve and clarified using monobasic sodium phosphate according to the process described by Sica and colleagues [43]. In this technique used in laboratory-scale fermentation experiments, the Maillard reaction occurs, but it has not been demonstrated to negatively impact growth efficiency and productivity. After clarification, the material was subjected to vacuum filtration with the aid of a vacuum pump and then filtered through 47 mm nylon membrane filters with pore sizes of 14 μm, 8 μm, and 0.45 μm. After removing the particles, the corn hydrolysate was diluted in water to reach a total reducing sugar concentration (TRS) of 130 g L$^{-1}$ and supplemented with 15 g L$^{-1}$ of $(NH_4)_2SO_4$, 3 g L$^{-1}$ of $KH_2PO_4$, and 0.5 g L$^{-1}$ of $MgSO_4 \cdot 7H_2O$. In addition to these nutrients, the medium was supplemented with the following trace elements: 4.5 mg L$^{-1}$ of $ZnSO_4 \cdot 7H_2O$; 0.3 mg L$^{-1}$ of $CoCl_2 \cdot 6H_2O$; 1 mg L$^{-1}$ of $MnCl_2 \cdot 4H_2O$; 0.3 mg L$^{-1}$ of $CuSO_4 \cdot 5H_2O$; 4.5 mg L$^{-1}$ of $CaCl_2 \cdot 2H_2O$; 3 mg L$^{-1}$ of $FeSO_4 \cdot 7H_2O$; 0.4 mg L$^{-1}$ of $NaMoO_4 \cdot 2H_2O$; 1 mg L$^{-1}$ of $H_3BO_3$; and 0.1 mg L$^{-1}$ of KI. Subsequently, the wort was sterilized in an autoclave at a temperature of 121 °C for 20 min and at 1 atm pressure [28].

### 4.2. Treatments

In cell growth, 3 treatments with 5 replications each were tested. The treatments carried out during cell multiplication (Table 2) aimed to demonstrate the behavior of the yeast at different oxygen concentrations in the medium.

**Table 2.** Treatments tested on cell growth.

| | Treatments |
|---|---|
| T1 | Cell growth at 0.5 volume of air per volume of wort per minute (v v$^{-1}$ min$^{-1}$) |
| T2 | Cell growth at 1.0 volume of air per volume of wort per minute (v v$^{-1}$ min$^{-1}$) |
| T3 | Cell growth at 1.5 volume of air per volume of wort per minute (v v$^{-1}$ min$^{-1}$) |

### 4.3. Yeast Acclimation and Rehydration

For the yeast to begin cellular growth in fed-batch mode, it was inoculated into 500 mL Erlenmeyer flasks containing 300 mL of YEPD culture medium with 50 g L$^{-1}$ of TRS for 4 h, in a shaker, Marconi® (Piracicaba, Brazil), model Ma 420, under agitation at 150 rpm, a temperature of 30 °C, and pH 4.5. At the end of the process, the yeast was separated from the culture medium by centrifugation (3925× *g* for 10 min) and inoculated into the cell growth reactor [40].

### 4.4. Fed-Batch Cell Growth

Cell growth began with 9 g L$^{-1}$ (dry mass) of *Saccharomyces cerevisiae*, strain Thermosacc Dry®, being inoculated into 2 L Erlenmeyer flasks containing 300 mL of corn

hydrolysate with a sugar concentration of 0.1 g L$^{-1}$. Cell growth was carried out at a temperature of 30 °C, pH 4.5, agitation at 150 rpm, and aeration according to the treatments in Table 2. To aerate the cultivation medium, a compressed air filter was used, which was injected into the reactor, and to determine the aeration that was being injected, the digital flow meter, Siargo® (Santa Clara, CA, USA), model Mf5706, (0–10 L), was used. Immediately after yeast inoculation, the reactor was fed to maintain the sugar concentration close to 0.1 g L$^{-1}$. Due to the final useful volume of the reactor being a limiting factor, wort containing a sugar concentration of 130 g L$^{-1}$ was used. A computer-controlled pumping system with a peristaltic pump was employed to add the wort to the reactor, and the feeding process was carried out over a 16 h period. Feeding started with a feed flow (Fs) of 0.21 mL min$^{-1}$, which was increased according to Equation (1). Pumping control was automated, through a computer, which was programmed following Equation (1) [19].

$$\text{Fs (t)} = \frac{\mu\ (\text{t})\ [\text{X (t) V (t)}]}{\text{Yx/s (t)}(\text{Sf} - \text{Sm})} \tag{1}$$

where:

Fs (t) is a function of the total biomass of yeast occupying a given volume, [X (t) V (t)];

μ (t): the specific growth rate;

Yx/s (t): cell yield coefficient;

Sf: the feed substrate concentration (g L$^{-1}$);

Sm: the substrate concentration at which the yield is the maximum (g L$^{-1}$).

The parameters used to calculate the feed flow were as follows: the specific growth rate (μ (t) = 0.1225 h$^{-1}$) and the cell yield coefficient (Yx/s (t) = 0.5 g cell/g of sugar).

### 4.5. Cell Viability of Yeast

Cell viability analysis was determined through the differential staining of cells in 0.1% methylene blue solution, using a Neubauer chamber and an optical microscope, Nikon® (Melville, NY, USA), model E 200, in a magnification of 400×, considering readings of live cells (transparent) and dead cells (blue), methodology described by Pierce [44].

### 4.6. Determination of Cell Biomass and Protein

Samples of centrifuged yeast were analyzed for dry biomass and protein using standard methods [45].

### 4.7. Determination of Total Sugars and Glycerol

Concentrations of glucose, fructose, and glycerol were determined from 1 mL samples of wort and wine. For this purpose, 0.25 μL of sample was injected into an ion chromatograph, from Metrohm®, model IC 930, using the following chromatographic system: chromatographic column, model Metrosep Carb 1—150/4.0; amperometric detector; and the eluent solution prepared was 200 mM sodium hydroxide and a flow rate of 1.0 mL min$^{-1}$. The column temperature was maintained at 35 °C, and the chromatographic run time was 9 min [46].

### 4.8. Determination of Alcohol Content

The alcohol content was determined in the wine samples using samples of yeast-free wine. These samples were distilled in an alcohol distiller, Tecnal® (Piracicaba, Brazil), model TE-010, and the alcohol content was analyzed using a densitometer model EDM 5000, Schmidt Haensch® (Berlin, Germany) [47].

### 4.9. Organic Acids

Wine samples were centrifuged to remove yeast, and the following parameters of the supernatant were analyzed by HPLC: succinic acid, lactic acid, and acetic acid. For this, 100 μL of the sample was diluted with 900 μL of ultrapure water and injected into an

HPX-87H ion-exchange column (Bio-Rad® located in Hercules, CA, USA) at 60 °C, with 5 mM $H_2SO_4$ as the mobile phase and a flow rate of 0.6 mL $min^{-1}$ [48].

### 4.10. Trealose

Trealose was extracted from yeast cells that were cooled and washed with cold 0.5 M trichloroacetic acid, and the trealose concentration was estimated using the anthrone method [49,50].

### 4.11. Statistical Analyses

Statistical analyses were performed using the R program v. 3.3.2 [51]. The experimental design used was a completely randomized design. First, the samples were submitted for ANOVA, and the Tukey test was used for mean comparison at a significance level of 5% ($p < 0.05$).

## 5. Conclusions

When growing yeast in a fed-batch system under different levels of aeration, it was found that yeasts grown in an aeration of 0.5 v $v^{-1}$ $min^{-1}$ in the reactor showed higher biomass formation and cell yield and a lower formation of organic acids. In addition, the yeasts exhibited desirable characteristics for use in alcoholic fermentation (high cell viability and high trehalose concentrations).

Adopting the aerobic yeast production process enables more efficient sugar utilization for biomass formation. Consequently, the surplus amount of sugar can be redirected to increase ethanol production and enhance the protein content of distillers' dried grains (DDG), thereby producing a co-product with a higher market value.

**Author Contributions:** Conceptualization, M.R.B.O. and A.S.B.; methodology, M.R.B.O. and A.S.B.; software, M.R.B.O.; validation, M.R.B.O., R.S.D., P.S., L.A.M. and A.S.B.; formal analysis, M.R.B.O., R.S.D. and A.S.B.; investigation, M.R.B.O., R.S.D., P.S., L.A.M., A.U.P. and T.M.F.; resources, A.S.B.; data curation, M.R.B.O., R.S.D. and A.S.B.; writing—original draft preparation, M.R.B.O.; writing—review and editing, M.R.B.O., R.S.D. and A.S.B.; visualization, M.R.B.O. and A.S.B.; supervision, M.R.B.O. and A.S.B.; project administration, M.R.B.O. and A.S.B.; funding acquisition, M.R.B.O. and A.S.B. All authors have read and agreed to the published version of the manuscript.

**Funding:** This research was funded in part by Coordenação de Aperfeiçoamento de Pessoal de Nível Superior-Brasil (CAPES), grant number 001.

**Data Availability Statement:** The raw data supporting the conclusions of this article will be made available by the authors on request.

**Conflicts of Interest:** The authors declare no conflicts of interest.

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
