# Peer review of "Evaluation of Aerobic Propagation of Yeasts as Additional Step in Production Process of Corn Ethanol"

_stresses, doi:10.3390/stresses4020025_

Round 1

Reviewer 1 Report

Comments and Suggestions for Authors

Dear authors of the manuscript "EVALUATION OF AEROBIC PROPAGATION OF YEASTS AS AN ADDITIONAL STEP IN THE PRODUCTION PROCESS OF CORN ETHANOL" I think your articule its very interesting, i included some coments on the manuscript, and i suggest to shorten the discussion.

Author Response

Thank you for your suggestions, the corrections and our comments are attached.

Reviewer 2 Report

Comments and Suggestions for Authors

The subject of ms is interesting but the purpose, hypothesis, and results are not clear.

The sentence in lines 35-39 requires clarification. Specifically, the statement “humans and animals cannot synthesize them” needs to be more precise. Humans and animals cannot synthesize certain amino acids, known as essential amino acids, which they must obtain from their diet. These amino acids are crucial for building proteins necessary for metabolic function and providing nitrogen.

Here is a suggested revision for clarity:

"Proteins are essential for metabolic function and serve as a source of nitrogen for humans and animals to build the structural and functional units necessary for life. However, humans and animals cannot synthesize certain essential amino acids, so they consume meat or other protein sources to meet these needs."

In lines 40-41, the reported range of 45% to 65% protein content in yeast biomass appears suspiciously high for the upper value. Most reliable sources suggest that the protein content of S. cerevisiae biomass typically falls within a more conservative range.

The sentence in lines 51-53, "The oxidation of sugars under aerobic conditions represents a more efficient energy pathway than anaerobiosis, yielding 30 molecules of ATP per molecule of glucose," might not be completely accurate. It would be more precise to say "around 30 ATP molecules" because the exact yield of ATP can vary slightly depending on the cell type and the efficiency of the mitochondrial machinery. Typically, the range is estimated to be between 30 to 32 ATP molecules per glucose molecule under aerobic conditions.

In its current form, the manuscript is written in a way that is difficult to understand and draw conclusions. It is not clear why the authors want to multiply the yeast. The strain used is a commercial strain that grows in high gravity conditions and is used to preferentially produce ethanol efficiently, not biomass. Commercial products are often either mixtures of strains or strains that are changed for production, for example, every month. Thus, using a commercial product might make it difficult to reproduce results. Therefore, it is difficult to compare the results to other well-characterized strains

The results start with a statement that they were treated under three different conditions (T1 to T3), which are described only later in the materials and methods section. This structure makes the manuscript difficult to read because the reader must refer to the materials and methods at the end to understand the results presented earlier.

The method of preparing the medium is complicated and difficult to understand. It is unclear why the medium prepared at the beginning contains 130 g/L of reducing sugars, but the inoculation medium (corn hydrolysate) contains only 0.10 g/L of reducing sugars, while 120 g/L is fed. Autoclaving the medium with sugar at 121°C can cause a Maillard reaction. The authors should comment on this fact in the materials and methods section.

Since the authors suggest yeast growth as an additional stage in ethanol production, they should present at least rough financial, economic, or ecological assumptions about how this process might affect costs and its potential profitability. Protein production by yeast is relatively inefficient and expensive, and using strains intended for ethanol production rather than biomass seems like an erroneous assumption for producing additional yeast biomass. The assumption of yeast cultivation on starch hydrolysate does not solve the waste problem; it only consumes raw material that could be used to produce ethanol.

Additionally, regarding protein production, the study should investigate whether available nitrogen is a limiting factor for the cultivation processes.

The observed results may result from increasing biomass and thus growing under aerobic conditions from T1 to T3 being the same for the first hours of cultivation. Only after the biomass has multiplied in cultures where oxygen transfer starts to be a limiting factor, anaerobic conditions occur faster in lower vvm, causing ethanol production. This seems to be confirmed by the result observed for T3, where the start of ethanol production was observed at the latest.

Comments on the Quality of English Language

Please proofread your text for clarity of the information.

Author Response

(The authors gave the same response as above.)

Round 2

Reviewer 2 Report

Comments and Suggestions for Authors

The ms was improved and in the present form may be published. Good luck:!